# Determination of Johnson–Cook Constitutive of 15-5 PH Steel Processed by Selective Laser Melting

**DOI:** 10.3390/ma16020800

**Published:** 2023-01-13

**Authors:** Xiaojing Zhang, Wenjin Yao, Xintao Zhu, Zhiming Hu, Wei Zhu, Hongxin Huang, Wenbin Li

**Affiliations:** 1School of Mechanical Engineering, Nanjing University of Science and Technology, Nanjing 210094, China; 2Aerocraft Technology Branch of Hunan Aerospace Co., Ltd., Changsha 410205, China

**Keywords:** selective laser melting, 15-5 PH steel, Johnson–Cook constitutive model, penetration, numerical simulations

## Abstract

To obtain a Johnson–Cook model of 15-5 PH steel formed by selective laser melting (SLM), and to determine the difference between the forging process, in this work, mechanical testing, penetration testing and numerical simulations were used to study 15-5 PH steel formed by SLM and forging. Finally, the Johnson–Cook model parameters and failure parameters of the 15-5 PH steel formed by SLM and forging were obtained. We found that the SLM process was beneficial for refining the grain size of 15-5 PH steel and for improving the mechanical properties of 15-5 PH steel, where the yield strength of its specimens increased by 13.1% compared with the forged specimens. The error between the numerical simulations and penetration tests was less than 10%, which verified the validity of the numerical model parameters. It was also found that the penetration ability and abrasion resistance of the SLM-shaped projectiles were slightly superior to those of the forged projectiles.

## 1. Introduction

To manage increasingly solid military targets, improving the penetration capability of warheads has become an urgent task [1]. Changing the shape and structure of the projectile or changing the material of the projectile are two common methods to improve the penetration ability of a projectile. Missile structures are formed using a mature design method, where the development of projectile materials mainly depends on the application of new materials and the development of processing technology. Currently, the traditional forging process is the preferred process for processing penetrating projectiles, and the impact of new processes (such as SLM) on the penetration performance of projectiles has been a topic of interest in recent years.

As a new processing technology, selective laser melting (SLM) technology uses a laser to melt and consolidate powder according to a planned path, stacking it layer by layer and forming it integrally [2]. Compared to traditional large-scale metal component manufacturing technologies such as forging, SLM has the advantages of not requiring large-scale supporting facilities and a small subsequent machining allowance. It also offers a high degree of flexibility and component design, with structural design no longer restricted by manufacturing technology [3,4]. Due to the extremely fast cooling speed, SLM parts can be endowed with a fine internal microstructure, endowing the parts with fine grains, uniform composition, dense structure, and excellent mechanical properties [5,6,7]. Raiyan Seede et al. [8] studied the density, microstructure, and mechanical properties of af9628 steel formed by SLM and found that the tensile yield strength of af9628 steel was 1.04–1.11 Gpa, the tensile strength was 1.31–1.43 GPa, and the elongation after fracture was 7.4–10.9%.

The parts formed through SLM manufacturing will show fine grains and excellent mechanical properties. Moreover, the strength of martensitic steel formed by SLM will be generally greater than that obtained by traditional preparation methods [9,10]. Tan [11] showed that SLM molding was conducive to the formation of martensite and the fine grain strengthening effect [12]. Yu et al. [13] showed that the remelting process resulted in a higher cooling rate and refined grain structure in 17-4 PH made by laser direct metal deposition. Jeong-Rim Leea et al. [14] showed that the average grain sizes of 15-5 PH steel formed by laser melting and sintering and traditional 15-5 PH steel were 11 μm and 2–3 μm, respectively, and noted that SLM-formed 15-5 PH steel had a significantly smaller grain size, finer microstructure, and higher hardness than traditional 15-5 PH steel. Research by Nong et al. [15] showed that SLM-formed 15-5 PH steel could obtain a highly dense structure with very fine grains, improving its strength and elongation [16].

By studying the Johnson–Cook constitutive model parameters of various metal materials, researchers verified that the J-C constitutive model was suitable for large deformation in metal materials [17,18]. Wang [19] et al. divided the range of strain rate by magnitude, and when the strain rate was less than 10^−5^ s^−1^, it belonged to static deformation. When the strain rate was between 10^−5^ and 10^−1^ s^−1^, it belonged to quasi-static deformation, and when it was greater than 10^−1^ s^−1^, it belonged to dynamic deformation. The inertia and strain rate effects of static deformation and quasi-static deformation could be neglected. During dynamic deformation, the material showed an obvious inertia effect and strain rate effect [20]. Singh et al. [21] reported the J-C constitutive model parameters of low carbon steel according to compression and tensile tests. The results showed that the J-C constitutive model could well describe the deformation of low carbon steel.

Geng et al. [22] carried out tensile tests on 35CrMnSiA, f175, and DT300 steels and fitted their J-C constitutive model parameters. The Taylor bar test and numerical simulation results showed that the J-C constitutive model could be used to study the above materials. Xu et al. [23] carried out dynamic and static mechanical property tests of G31 steel and fitted the J-C constitutive parameters of G31 steel, and the numerical simulation results verified the accuracy of the J-C constitutive parameters. Wei et al. [24] fitted the J-C constitutive model of 38crsi high-strength steel with the tensile test results, which showed that the J-C model with a modified temperature term coefficient could be used for large deformation analysis of 38crsi materials and structures under impact loading. Niu et al. [25] studied the mechanical properties of 30CrMnSiNi2A steel in the temperature range of 30–700 °C with a strain rate range of 3000–10,000 s^−1^. The results showed that the material had obvious temperature sensitivity at 300 °C, and the flow stress decreased significantly when the temperature continued to increase. The modified J-C constitutive model could describe the deformation of the material well. The applications of the J-C constitutive model in metal materials are considered very mature. In this work, the J-C constitutive models of SLM and forged 15-5 PH steel were obtained through a series of experiments.

Currently, research on SLM-formed 15-5 PH steel is not comprehensive, with no systematic mechanical property and penetration tests. Differing from previous studies, to systematically study the influence of the mechanical properties and penetration properties of SLM-formed and forging-formed 15-5PH steel, quasi-static compression, SHPB, notch tensile, quasi-static tensile, high-temperature compression, and high-temperature tensile tests were carried out on the specimens of the two materials. Subsequently, the Johnson–Cook (J-C) constitutive model parameters and failure parameters of the forged and SLM-formed 15-5 PH steel were obtained by mechanical test fitting. The parameters were brought into the numerical model to simulate the penetration at different velocities, and finally, the reliability of the parameters was verified by the penetration test results.

## 2. Johnson–Cook Constitutive

The J-C constitutive model was proposed by Johnson and Cook [26] in 1983, and can represent the strength behavior of materials, typically metals, subjected to large strains, high strain rates, and high temperatures. This type of behavior could cause problems of intense impulsive loading due to high velocity impacts, and with this model, the yield stress would vary depending on the strain, strain rate, and temperature. The effects of strain hardening, strain rate hardening, and temperature softening on the mechanical properties of the materials were comprehensively considered, according to the following expression:(1)$$\sigma_{e}=\left(A+B\varepsilon_{e,p}^{n}\right)\left(1+Cln\frac{\dot{\varepsilon_{e,p}}}{\dot{\varepsilon_{0}}}\right)\left[1-{\left(\frac{T-T_{r}}{T_{m}-T_{r}}\right)}^{m}\right]$$
where $\sigma_{e}$ is the flow stress, $\varepsilon_{e,p}$ is the effective plastic strain, $\dot{\varepsilon_{e,p}}$ and $\dot{\varepsilon_{0}}$ are the effective plastic strain rate and the reference strain rate, respectively, and T, $T_{r}$, and $T_{m}$ are the test temperature, the reference temperature, and the melting point of the material, respectively. In addition, *A*, *B*, *C*, *n*, and *m* denote the undetermined parameters of the model, whose values depend on the properties of the material itself, where *A* is the yield strength under a reference temperature and reference strain rate, *B* and *n* are the strain hardening coefficients, *C* is the strain rate sensitivity coefficient, and *m* is the temperature sensitivity coefficient.

The J-C failure model [27] was proposed by Johnson and Cook in 1985, and comprehensively considers the effects of stress triaxiality, strain rate, and temperature on material failure, with the following expression:(2)$$\varepsilon_{f}=\left[D_{1}+D_{2}exp\left(D_{3}\sigma^{*}\right)\right]\left(1+D_{4}ln\frac{\dot{\varepsilon_{e,p}}}{\dot{\varepsilon_{0}}}\right)\left[1+D_{5}\frac{T-T_{r}}{T_{m}-T_{r}}\right]$$
where $\varepsilon_{f}$ is the failure strain, $D_{1}–D_{5}$ are the failure parameters of the material, and the dimensionless compressive stress ratio is given by
$$\sigma^{*}=p/\bar{\sigma }=-\eta$$
where p is the hydrostatic pressure, $\bar{\sigma }$ is the equivalent stress, η is the stress triaxial degree [28], and $\dot{\varepsilon_{e,p}}$, $\dot{\varepsilon_{0}}$, T, $T_{r}$, and $T_{m}$ have the same meaning as in Equation (1).

Material element damage could be defined by the J-C failure model as follows:(3)$$D=\sum \frac{\Delta \varepsilon_{p}}{\varepsilon_{f}}$$
where D is the damage parameter of the material element, and its value varies from 0 to 1 (initially, D = 0, and when D = 1, the material fails), $\Delta \varepsilon_{p}$ is the plastic strain increment of a one time step, and $\varepsilon_{f}$ is the failure strain under the stress state, strain rate, and temperature of the current time step.

The parameters in the J-C constitutive model were obtained from the corresponding mechanical tests; *A*, *B*, and *n* were obtained by quasi-static test fitting, C was obtained by SHPB test fitting, N was obtained by high-temperature compression testing, *D*_1_–*D*_3_ were obtained by notch tensile test fitting, *D*_4_ was obtained by quasi-static tensile test fitting, and *D*_5_ was obtained by high-temperature tensile testing.

## 3. Mechanical Testing

### 3.1. Preparation of the 15-5 PH Steel

The chemical composition of the forged 15-5 PH steel is presented in Table 1, where the heat treatment process consisted of common solid solution treatment + aging treatment. According to the research of Qiu et al. [29], solid solution treatment consisted of 1040 °C heat preservation for 55 min and 20–40 °C oil cooling, while aging treatment consisted of heat preservation at 465 °C for 4 h and water cooling at 65 °C. The hardness of the forging after heat treatment was measured to be 44 HRC.

The chemical composition of the powder required for molding 15-5 PH steel by SLM is shown in Table 2, while the particle size distribution of the powder obtained by the sieve method is shown in Table 3.

The heat treatment process of the SLM-shaped specimen involved stress relief annealing. The specific method consisted of maintaining 482 °C for 4 h, followed by cooling in the furnace. The scanning strategy utilized a strip-shaped partition, with a partition width of 5 mm, S-shaped bidirectional wiring, and 67° interlayer rotation. The SLM scanning strategy schematic diagram, the physical surface of the molded part, and the SLM process diagram are shown in Figure 1.

The measured SLM was 7.779 g/cm^3^, while the density of the forged 15-5 PH steel was 7.850 g/cm^3^. The density similarity between the two was 99.1%.

### 3.2. Quasi-Static Compression Test

#### 3.2.1. Quasi-Static Compression Principle and Specimen Design

Compression testing applies axial pressure to the specimen and measures the strength and plasticity of the material during its deformation and fracture [30]. The CSS-44100 testing machine used in the test had a maximum test pressure of 100 KN, and the clamping of the test piece is shown in Figure 2.

The strength of the test piece was high; thus, the upper and lower end faces of the test piece were equipped with high hardness cushion blocks to protect the test machine platform. This ensured that the loading forces of the cushion block, test piece, and testing machine were coaxial. We applied a thin layer of grease on the contact surface between the test piece and the cushion block to reduce friction. Moreover, a compression extensometer was installed on the cushion block. The hardness of the cushion block was much higher than that of the test piece, and we considered that there was no deformation. Therefore, the displacement of the cushion block measured by the extensometer was the axial deformation of the test piece, which ensured the accuracy of the compression test.

We collected data on the load of the testing machine and the corresponding deformation during the test and calculated the engineering stress-strain curve of the test piece as follows:(4)$$\{\begin{array}{c} \sigma =\frac{4F}{\pi d_{0}^{2}}\\ \varepsilon =\frac{\Delta l}{l_{0}} \end{array}$$
where F is the load of the testing machine, *d*_0_ is the initial diameter of the test piece, $l_{0}$ is the initial length of the test piece, Δl is the deformation of the test piece, namely, the data recorded by the extensometer, and σ and ε are the engineering stress and engineering strain of the test piece, respectively.

During the quasi-static compression process, the diameter of the specimen will increase. It would be unreasonable to always use the initial diameter *d*_0_ in Equation (4). It was necessary to replace *d*_0_ with the instantaneous diameter *d* to correct Equation (4). Assuming that the volume of the specimen remained unchanged during deformation, Equation (5) was obtained:(5)$$\frac{\pi d^{2}}{4}l=\frac{\pi d_{0}^{2}}{4}l_{0}$$
where *l* is the instantaneous length of the test piece, and $l=l_{0}-\Delta l$.

The true stress-strain curve of the test piece was obtained as follows [31]:(6)$$\{\begin{array}{c} \sigma_{T}=\frac{4F}{\pi d_{0}^{2}}\cdot \frac{l}{l_{0}}=\sigma \left(1-\varepsilon \right)\\ \varepsilon_{T}=\int_{l}^{l_{0}}\frac{dl}{l}=-ln\left(1-\varepsilon \right) \end{array}$$
where $\sigma_{T}$ and $\varepsilon_{T}$ are the true stress and true strain, respectively.

During the compression test, there was great friction force at the end face of the sample, which hindered the transverse deformation of the sample, causing the sample to have a waist drum shape, affecting the accuracy of the test results, where the smaller the height to diameter ratio L/d, the greater the influence of the end friction on the test results. The ratio could be appropriately increased to reduce the impact of end friction; however, an excessive ratio would lead to longitudinal instability [30]. According to the national standard requirement of 1 ≤ L/d ≤ 2, the nominal size of the designed test piece was Ø5 × 6 mm. The loading speed V of the testing machine was set to 0.36 mm/min, and the strain rate calculated according to Equation (7) [32] was 1 × 10^−3^ s^−1^.
(7)ε˙=ΔεΔt=ΔLLΔt=ΔLΔtL=vL

#### 3.2.2. Quasi-Static Compression Results and Analysis

The test piece before and after the test had a waist drum shape, as shown in Figure 3.

The real stress-strain curves of SLM and forged 15-5 PH steel specimens were obtained by processing the load displacement data obtained from the universal testing machine using Equations (4)–(6), as shown in Figure 4.

As shown in Figure 4, the real stress-strain curves of SLM and forged 15-5 PH steel specimens have good repeatability. Figure 5 shows that the true stress-strain curve of the SLM-formed 15-5 PH test piece had a micro-segment platform after the yield point. From the perspective of forming technology, the reason for this phenomenon was possibly because some of the powder materials were not fully melted and consolidated during SLM forming, resulting in a small number of pores in the formed specimen. During the forging process, the specimen was upset and compacted many times, and the interior of the specimen was dense. Therefore, the pores in the SLM-shaped specimen were densely compressed during the compression process and would be concentrated near the yield point and then compacted. For macroscopic performance, the strain increased, while the stress was almost unchanged. This also confirmed that the SLM 3D printing mentioned above caused the density to be slightly less than 100%. Moreover, the work hardening effect of the SLM-shaped specimen was more obvious than that of the forged specimen, and the resistance to plastic deformation was stronger.

Taking the true stress value corresponding to the plastic strain of 0.2% as the yield strength value, we found that the quasi-static yield strength values of the SLM and forged 15-5 PH steel at room temperature were 1464 and 1295 MPa, respectively. The yield strength of the SLM-shaped specimen was 13.1% higher than that of the forged specimen.

### 3.3. Split Hopkinson Bar (SHPB) Test

#### 3.3.1. SHPB Test Principle and Specimen Design

During the process of penetration, the strain rate borne by the projectile will be between 10^3^ and 10^4^ s^−1^ [23]. The mechanical response of the 15-5 PH steel under dynamic impact significantly differed from that under quasi-static impact [33]. Therefore, it was necessary to carry out dynamic mechanical tests to obtain the dynamic mechanical properties. Split Hopkinson bar (SHPB) [20] is the most commonly used test device for dynamic compression testing with a strain rate between 10 and 10^4^ s^−1^.

The SHPB device used in this test is shown in Figure 6. The diameter of the bullet, the incident rod, and the transmission rod are all 14.5 mm, the bullet is 200 mm long, the incident rod is 2000 mm long, and the transmission rod is 1500 mm long. The rod is made of high-strength spring steel, with an elastic modulus of 210 GPa and a density of 7.85 g/cm^3^.

Of note, the left and right end faces of the test piece were end face 1 and end face 2, and their respective displacements were $U_{1}$ and $U_{2}$. According to the linear superposition principle of linear elastic waves, we have
(8)$$\{\begin{array}{c} U_{1}=c_{0}\int_{0}^{t}\left(\varepsilon_{i}-\varepsilon_{r}\right)d\tau \\ U_{2}=c_{0}\int_{0}^{t}\varepsilon_{t}d\tau \end{array}$$
where $\varepsilon_{i}$, $\varepsilon_{r}$, and $\varepsilon_{t}$ denote the strain of the incident wave, reflected wave, and transmitted wave measured by the resistance strain gauge, respectively, and c0=Eρ is the one-dimensional stress wave velocity of the rod, where *E* and ρ are the elastic modulus and density of the rod, respectively. The average strain of the test piece is given by
(9)$$\varepsilon \left(t\right)=\frac{U_{1}-U_{2}}{l_{0}}=\frac{c_{0}}{l_{0}}\int_{0}^{t}\left(\varepsilon_{i}-\varepsilon_{r}-\varepsilon_{t}\right)d\tau$$
where $l_{0}$ is the initial length of the test piece, and the average strain rate of the test piece can be obtained by deriving the time from Equation (6), as follows:(10)$$\dot{\varepsilon }=\frac{c_{0}}{l_{0}}\left(\varepsilon_{i}-\varepsilon_{r}-\varepsilon_{t}\right)$$

We could assume that the forces on end face 1 and end face 2 are $F_{1}$ and $F_{2}$; then, we have
(11)$$\{\begin{array}{c} F_{1}=AE\left(\varepsilon_{i}+\varepsilon_{r}\right)\\ F_{2}=AE\varepsilon_{t} \end{array}$$
where *A* is the cross-sectional area of the rod, and the average stress in the test piece is given by
(12)$$\sigma =\frac{\left(F_{1}+F_{2}\right)}{2A_{0}}=\frac{AE\left(\varepsilon_{i}+\varepsilon_{r}+\varepsilon_{t}\right)}{2A_{0}}$$
where $A_{0}$ is the initial cross-sectional area of the test piece. When the forces on both ends of the test piece were balanced, we considered that the uniform stress and deformation processes occurred in the test piece. The average stress represented the one-dimensional stress wave state in the material, according to
(13)$$\varepsilon_{i}+\varepsilon_{r}=\varepsilon_{t}$$

Substituting Equation (13) into Equations (6), (7), and (9), we obtain
(14)$$\{\begin{array}{c} \sigma =\frac{AE}{A_{0}}\varepsilon_{t}\\ \varepsilon =-\frac{2c_{0}}{l_{0}}\int_{0}^{t}\varepsilon_{r}d\tau \\ \dot{\varepsilon }=-\frac{2c_{0}}{l_{0}}\varepsilon_{r} \end{array}$$
where σ, ε, and $\dot{\varepsilon }$ are the engineering stress, engineering strain, and strain rate of the test piece, respectively, and the engineering stress-strain curve of the test piece was obtained when the strain rate was $\dot{\varepsilon }$ [20].

The size of the specimen had to meet two basic assumptions of SHPB experimental technology, i.e., the one-dimensional stress wave hypothesis and the assumption that the stress and strain in the specimen were uniformly distributed along the length of the specimen [20]. The aspect ratio recommended by the American Metal Society (ASM) is 0.5-1 [34]. Due to the high strength of 15-5 PH steel, to make the specimen easily yield at a high strain rate, the diameter of the specimen had to be appropriately reduced. Thus, the nominal size of the specimen was determined to be ∅4×3 mm, according to the specimen size suggested by Zhang et al. [35].

#### 3.3.2. SHPB Test Results and Analysis

The SLM and forged test pieces were subjected to dynamic compression loading for four groups of different strain rates, where the real stress-strain curve was obtained by the two-wave method, as shown in Figure 7.

As shown in Figure 7a, under a strain rate of 1300 s^−1^, the dynamic yield strength of the 15-5 PH steel formed by SLM was about 14.6% higher than the quasi-static yield strength, and the strain rate strengthening effect was obvious. As shown in Figure 7b, no fracture occurred at a strain rate of 5600 s^−1^. This was possibly because the heat accumulation of the forged specimen was faster, and the thermal softening effect during the initial stage of plastic strain exceeded strain strengthening. The entire plastic strain stage reflected thermal softening.

It can be seen in Figure 7 that as the strain rate increases, the yield strain of the material decreases, and the yield stress of the material increases slightly. This feature of the forging material is obvious, while for the SLM material, the change is less evident.

The relationship between compressive yield strength and strain rate of the SLM and forged 15-5PH steel under quasi-static conditions, as presented in Table 4, is shown in Figure 8.

As shown in Figure 7 and Figure 8, the yield strength values of the SLM and forged 15-5 PH steel increased with increasing strain rate, reflecting the strain rate effect. The dynamic yield strength of the SLM-formed 15-5 PH steel was higher than that of the forged 15-5 PH steel, indicating that the SLM forming process was helpful for improving the dynamic mechanical properties of 15-5 PH steel.

There was no obvious yield point in the result curves of the SHPB test. The true stress value corresponding to 0.2% plastic strain was selected as the yield strength, and the results are presented in Table 4.

### 3.4. Notch Tensile Test

#### 3.4.1. Notch Tensile Test Process and Specimen Design

To reflect the complex stress state of the material under stress, the stress triaxial degree *η* was cited as the complex stress state parameter, and it can be expressed as follows [36]:(15)$$\eta =\frac{\sigma_{m}}{\bar{\sigma }}=\frac{\frac{\sigma_{1}+\sigma_{2}+\sigma_{3}}{3}}{\sqrt{\frac{{\left(\sigma_{1}-\sigma_{2}\right)}^{2}+{\left(\sigma_{2}-\sigma_{3}\right)}^{2}+{\left(\sigma_{3}-\sigma_{1}\right)}^{2}}{2}}}$$
where $\sigma_{1}$, $\sigma_{2}$, $\sigma_{3}$, $\sigma_{m}$, and $\bar{\sigma }$ represent the three principal stresses, average stress, and equivalent stress, respectively.

The failure strain of the material was related to the stress state, and the stress state generally showed triaxial η characterization. According to the research by Bridgman [37], the relationship between the initial stress triaxiality at the center of the notched specimen, notch radius, and notch section diameter could be expressed by the following formula:(16)$$\eta_{0}=\frac{1}{3}+\ln\left(1+\frac{D}{4R}\right)$$
where $\eta_{0}$, *D*, and *R* indicate the initial stress triaxiality, notch section diameter, and the notch radius, respectively.

To obtain a relationship between the stress triaxiality and failure strain of the SLM and forged 15-5 PH steel, notch tensile tests were carried out at the reference temperature and strain rate. The nominal size of the notch section diameter D was 5 mm, while the notch radius R values were 2.5 mm, 5 mm, and 7.5 mm. The dimensions of the notch test piece and finished product are shown in Figure 9.

A UTM5105-G electronic universal testing machine was used for the notch tensile test, where the maximum test force was 100 KN. The tensile rate of the test was 1.8 mm/min, and the strain rate was 1 × 10^−3^ s^−1^. The test piece was stretched to the end of fracture.

#### 3.4.2. Notch Tensile Test Results and Analysis

It is often assumed that the volume will be constant during tensile testing, with uniform strain at the fracture section. According to research by Børvi et al. [38], the failure strain of the test piece can be expressed as follows:(17)$$\varepsilon_{f}=\ln\frac{A_{0}}{A_{f}}$$
where $\varepsilon_{f}$, $A_{0}$, and $A_{f}$ are the failure strain, initial cross-sectional area, and fracture cross-sectional area of the test piece, respectively, and the formula can be further simplified as follows:(18)$$\varepsilon_{f}=2\ln\frac{d_{0}}{d_{f}}$$
where $d_{0}$ and $d_{f}$ are the initial section diameter and fracture diameter of the test piece, respectively.

Because the nominal size of the cylindrical specimen was 5 mm for the quasi-static tensile test with a strain rate of 1 × 10^−3^ s^−1^, the *R* of the notch specimen could be regarded as infinite, and the test results were included. The failure strain of the notch tensile test could be obtained from Equation (18), as shown in Table 5 and Table 6.

As shown in Table 5 and Table 6, for the 15-5 PH steel specimens under the same preparation conditions, the fracture diameter of the specimens was inversely proportional to the change trend of the notch radius, while the failure strain was proportional to the change trend of the notch radius. Application of the SLM manufacturing process increased the failure strain of the specimen, which was significantly higher than that of the specimen fabricated via the forging process.

### 3.5. Quasi-Static Tensile Tests under Different Strain Rates

#### 3.5.1. Principle of the Quasi-Static Tensile Test and Design of the Specimens

The tensile test is a mechanical property test of a standard tensile specimen under the action of static axial tensile force at a specific tensile speed until fracture, which involves continuously recording the force and elongation during the tensile process and obtaining its strength criterion and plasticity criterion [39]. Figure 10 shows the dimensions and outline of the test piece.

A UTM5105-G electronic universal testing machine was used for the test, where the maximum test force was 100 kN. The test clamping state is shown in Figure 11.

The tensile rates of the testing machine were set to 0.9, 1.8, and 18 mm/min, and the strain rates were 5 × 10^−4^ s^−1^, 1 × 10^−3^ s^−1^, and 1 × 10^−2^ s^−1^, respectively. The test was not terminated until the test piece was broken.

#### 3.5.2. Results and Analysis of the Quasi-Static Tensile Test

The quasi-static tensile specimens under different strain rates consisted of round bar specimens, which could be regarded as notch tensile specimens with infinite notch radius, where the initial stress triaxiality was 0.333. The tensile test results were substituted into Equation (15) to calculate the failure strain, as shown in Table 7.

As shown in Table 7, the fracture diameter of the SLM-shaped specimen was smaller than that of the forged specimen, and its elongation after fracture was higher, which indicated that the SLM-shaped specimen had better ductility and higher plasticity. The failure strain data in Table 7 are shown in Figure 12. With an increase in strain rate, the failure strain of the SLM and forged specimens increased. At the same strain rate, the failure strain of the SLM specimens was larger than that of the forged specimens.

The stress state of the specimens in the tensile test differed from that in the compression test. Equations (4)–(6) were derived again to obtain the true stress-strain formula of the tensile test (19) [23]:(19)$$\{\begin{array}{c} \sigma_{T}=\frac{4F}{\pi d_{0}^{2}}\cdot \frac{l}{l_{0}}=\sigma \left(1+\varepsilon \right)\\ \varepsilon_{T}=\int_{l_{0}}^{l}\frac{dl}{l}=ln\left(1+\varepsilon \right) \end{array}$$
where the parameters have the same definitions as in Equations (1)–(3), and the expression of the instantaneous length of the test piece became $l=\Delta l+l_{0}$.

After modifying the elastic section of the tensile curve, the true stress-strain curve of the quasi-static tensile test under different strain rates was obtained using Equation (19), as shown in Figure 13.

We used the peak value of the curve in Figure 13 to obtain the tensile strength of the 15-5 PH steel, as shown in Table 8.

As shown in Table 8, under the quasi-static condition, the tensile strength of the SLM and forged 15-5 PH steel increased with an increase in strain rate, and the tensile test also showed the strain rate sensitivity. The tensile strength of the SLM-shaped specimen increased by about 16.4% compared to that of the forged specimen. The fracture morphologies of the SLM and forged 15-5 PH steel tensile test pieces are shown in Figure 14.

Figure 14 shows an obvious local necking phenomenon at the fracture of the SLM-shaped specimen, which produced obvious plastic deformation, where the smaller the strain rate, the more obvious the necking phenomenon. The fracture diameter of the forged specimen slightly decreased, with no obvious necking phenomenon.

For more careful observations, a field emission scanning electron microscope (SEM) was used to photograph and obtain microscopic fracture images of the test piece under a strain rate of 0.001 s^−1^, as shown in Figure 15.

Figure 15 shows the fracture characteristics of the specimen section on a more microscopic level. As shown in Figure 15a, the fracture section of the SLM-shaped specimen contained three parts: the fiber region, radiation region, and shear lip. The radiation area had obvious ridge-like and radial stripe-like features. The shear lip was located at the bright edge of the fracture surface, and its surface was smooth, with diameter shrinkage in the tensile stress direction. This was a typical feature of ductile fracture. We determined that the SLM-shaped specimen showed ductile fracture. As shown in Figure 15b, the cross-section of the forged specimen was relatively flush, the shear lip was very thin, and the shear lip was nearly perpendicular to the cross-section, with obvious brittle fracture characteristics. No obvious necking phenomenon was observed at the fracture of the forged specimen, and the fracture was perpendicular to the tensile stress; the fracture surface was flush, and the edge had a small and thin shear lip, indicating typical brittle fracture morphology. The shear lip thickness of the forged specimen was narrower than that of the SLM specimen, which also showed that SLM technology improved the plasticity of 15-5 PH steel.

### 3.6. High-Temperature Compression Test

During the process of high-speed penetration, the penetrating projectile will experience violent friction with the target. With time of the order of microseconds, the heat will not sufficiently escape, causing the temperature of the projectile and its surrounding objects to increase and the material falling off the target to become molten, causing it to finally cool and bond to the projectile. Thus, the penetrating projectile will experience a high-temperature environment during the process of action, and the mechanical properties of metal materials at high temperatures will be different from those at room temperature. Therefore, it was necessary to carry out high-temperature compression and high-temperature tensile mechanical tests to obtain the effect of the SLM forming process on the high-temperature mechanical properties of 15-5 PH steel.

According to the results of the high-temperature compression mechanical test conducted by Niu et al. [25] on the ultra-high strength steel 30CrMnSiNi2A, the yield strength of the ultra-high strength steel 30CrMnSiNi2A decreased by about 17% at a high temperature of 300 °C, and the difference started to become obvious at normal temperatures. At 500 °C, the mechanical properties decreased more seriously, reaching about 40%. Due to the higher set temperature of the high-temperature test, the larger the fluctuation range of the final test temperature, the longer the time required for the temperature to increase in the high-temperature furnace, and the long-time heating process could also damage the original heat treatment state of the specimen. Therefore, according to the temperature selection of Niu, 400 °C was taken as the test temperature for the high-temperature test.

#### 3.6.1. High-Temperature Compression Test Process

The clamping diagram showing the high-temperature compression test device and test piece is shown in Figure 16, where the test machine model was MTS370.10. We applied a thin layer of high temperature resistant grease on both ends of the test piece to reduce friction. Then, the test piece was coaxially clamped and gradually heated to 400 °C. After reaching 400 °C, the test was conducted after holding for 10 min [30]. We set the loading rate to 0.36 mm/min, where the strain rate was 1 × 10^−3^ s^−1^.

#### 3.6.2. High-Temperature Compression Test Results

The data processing of the high-temperature compression test was the same as that for quasi-static compression, and the true stress-strain curve of the SLM and forged 15-5 PH steel at 400 °C is shown in Figure 17.

The true stress-strain curve in Figure 17 had no obvious yield point; thus, the true stress value corresponding to 0.2% of the plastic strain was taken as the compression yield strength. The compressive yield strengths of the SLM and forged 15-5 PH steel at 400 °C were 1142 and 1007 MPa, respectively. We observed that the compressive yield strength of the SLM-shaped specimen was slightly higher than that of the forged specimen at 400 °C, only increasing by 13.4%. By comparing the yield stress at room temperature discussed in Section 3.2, we found that the yield stress of the SLM-shaped specimens decreased by 22.0% from room temperature to 400 °C, and the yield stress of the forged specimens decreased by 22.2%. We also observed that the temperature softening effect had a greater impact on the SLM-shaped specimen.

### 3.7. High-Temperature Tensile Test

#### 3.7.1. High-Temperature Tensile Test Process

Considering the clamping requirements of the high-temperature tensile testing machine, only the cylindrical parts at both ends were processed into threads without changing the size of the tensile specimens at room temperature, as shown in Figure 18c. The model of the high-temperature tensile testing machine was Zwick Roell Z100, and the clamping diagram of the high-temperature tensile test device and test piece is shown in Figure 18.

After the specimen was clamped, the high-temperature furnace was gradually heated to 400 °C. During the heating process, the temperature of the test piece did not exceed the upper allowable deviation limit of 403 °C. After reaching 400 °C, the tensile test was started after holding for 10 min [30]. The test loading rate was set at 1.8 mm/min, and the strain rate was 1 × 10^−3^ s^−1^.

#### 3.7.2. High-Temperature Quasi-Static Tensile Test Results

The fracture data of the tensile specimen at 400 °C were substituted into Equation (15) to calculate the failure strain of the specimen, and the results are shown in Table 9.

Table 9 shows that the failure strain of the SLM-formed specimen was less than that of the forged specimen at 400 °C, indicating that the tensile failure capacity of the SLM-formed specimen at a high temperature was weaker than that of the forged specimen. Compared to the quasi-static tensile test results at room temperature, the failure strain of the SLM specimens decreased by 29.6%, while the failure strain of the forged specimens increased by 25.1%. The failure strain of the conventionally forged materials increased with increasing temperature [23]. The forged 15-5 PH steel in this work conformed to this characteristic; however, the SLM-formed 15-5 PH steel showed a downward trend, which was affected by the forming mode. We observed that the temperature softening effect had a greater impact on the SLM-shaped specimen. Figure 19 shows the true stress-strain curves of the SLM and forged 15-5 PH steel at 400 °C.

The true stress-strain curve in Figure 19 had no obvious yield point. Taking the true stress value corresponding to the plastic strain of 0.2% as the tensile yield strength, the tensile yield strengths of the SLM and forged 15-5 PH steel at 400 °C were 998 and 1141 MPa, respectively. The tensile yield strength of the SLM specimens was 12.5% lower than that of the forged specimens. Compared to the quasi-static tensile test results at room temperature, the tensile yield strength of the SLM specimens decreased by 40.2%, while the tensile yield strength of the forged specimens decreased by 19.1%. This indicated that the tensile strength of the SLM-formed 15-5 PH steel was more affected by temperature.

Lu et al. [40] studied the thermal effect of a kinetic energy earth penetrating projectile penetrating reinforced concrete. When the ovoid penetrator penetrated the reinforced concrete at a speed of 1100 m/s, the maximum temperature of its warhead was 97 °C. The temperature change of the penetrating projectile under the condition of normal velocity penetration was not large enough to have a significant impact on the strength of the projectile. The projectile body mainly bore the compressive stress during the penetration process; thus, the tensile properties of the 15-5 PH steel formed by SLM at 400 °C decreased, and the impact on the penetration process was limited.

## 4. Determination of J-C Model for 15-5 PH Steel

### 4.1. Fitting Method and Results of the J-C Constitutive Parameters

#### 4.1.1. Fitting Parameters A, B, and n

Under the reference strain rate and reference temperature, the contents of the last two brackets on the right side of Equation (1) were both 1, which could obtain the following:(20)$$\sigma_{e}=A+B\varepsilon_{e,p}^{n}$$

After changing Equation (20), we obtain
(21)$$\ln\left(\sigma_{e}-A\right)=lnB+nln\varepsilon_{e,p}$$
where *A* is the quasi-static yield strength of 15-5 PH. The plastic deformation section of the quasi-static compression curve $\sigma_{T}-\varepsilon_{T}$ was transformed into curve $\ln\left(\sigma_{e}-A\right)$-$ln\varepsilon_{e,p}$, and the intercept of the obtained straight line was *lnB*, and the slope was *n*. Parameters *B* and *n* were obtained, and parameters *A*, *B*, and *n* were obtained by means of the quasi-static compression test results [31], as shown in Table 10.

The fitting of *A*, *B*, and *n* is shown in Figure 20, where the degree of coincidence between the fitting curve and test curve indicated a good fitting effect of parameters *A*, *B* and *n*.

#### 4.1.2. Fitting Parameter C

At the reference temperature, Equation (1) could be simplified as follows:(22)$$\sigma_{e}=\left(A+B\varepsilon_{e,p}^{n}\right)\left(1+Cln\frac{\dot{\varepsilon_{e,p}}}{\dot{\varepsilon_{0}}}\right)$$

Taking the plastic strain $\varepsilon_{e,p}$ in Equation (22) as 0, we could obtain
(23)$$\sigma_{yd}=A\left(1+Cln\frac{\dot{\varepsilon_{e,p}}}{\dot{\varepsilon_{0}}}\right)$$

For analysis based on SHPB tests, the C parameters for the SLM and forged 15-5 PH steel were 0.01368 and 0.01687, respectively. We compared the fitting curve obtained by Equation (23) with the test data, and the results are shown in Figure 21.

#### 4.1.3. Fitting Parameter m

Under the reference strain rate, Equation (1) could be simplified as follows:(24)$$\sigma_{e}=\left(A+B\varepsilon_{e,p}^{n}\right)\left[1-{\left(\frac{T-T_{r}}{T_{m}-T_{r}}\right)}^{m}\right]$$
where *A*, *B,* and *n* used the fitting results, the reference temperature $T_{r}$ was 298 K, and the material melting point $T_{m}$ could be calculated by the calculation method of the special steel in the empirical formula of steel liquidus [41], taking the average mass fraction. The empirical formula was as follows:(25)$$\begin{array}{ll} t_{1}=1536-(0.1 & +83.9\omega_{\left[C\right]}+10\omega_{\left[C\right]}^{2}+12.6\omega_{\left[Si\right]}+5.4\omega_{\left[Mn\right]}+4.6\omega_{\left[Cu\right]}+5.1\omega_{\left[Ni\right]}+1.5\omega_{\left[Cr\right]}-33\omega_{\left[Mo\right]}\\ \; & -30\omega_{\left[P\right]}-37\omega_{\left[S\right]}-9.5\omega_{\left[Nb\right]})-6 \end{array}$$
where $t_{1}$ is the liquidus temperature of steel (°C), $\omega_{\left[C\right]}$ is the mass fraction of carbon element in the steel (unit%), as well as for other elements such as silicon. The liquidus temperature $t_{1}$ of 15-5 PH steel formed by SLM was calculated as 1467.20 °C, and the melting point $T_{m}$ was calculated as 1740.35 K. The liquidus temperature $t_{1}$ of the forged 15-5 PH steel was calculated as 1455.96 °C, and the melting point $T_{m}$ was calculated as 1729.11 K.

According to the high temperature compression test results, the *m* parameter values of the SLM and forged 15-5 PH steel were 1.1242 and 1.1230, respectively. We compared the fitting curve obtained by Equation (25) with the test data, and the results are shown in Figure 22.

### 4.2. Parameter Fitting Method and Results of the J-C Failure Model

#### 4.2.1. Fitting Parameters D_1_, D_2_, and D_3_

At a reference temperature (298 K) and reference strain rate (1 × 10^−3^ s^−1^), Equation (2) converted to the following form:(26)$$\varepsilon_{f}=D_{1}+D_{2}exp\left(D_{3}\sigma^{*}\right)$$

Equation (26) represents the relationship between material failure strain and stress triaxiality under the reference temperature and reference strain rate. According to the notch tensile test results, failure parameters *D*_1_, *D*_2_, and *D*_3_ of the SLM and forged 15-5 PH steel were obtained, as shown in Table 11.

The comparison between the fitting curve obtained according to Equation (26) and the test data points is shown in Figure 23. As shown in Figure 23, the J-C failure curve fitting effect of the SLM and forged test pieces was very good, indicating that parameters *D*_1_, *D*_2_, and *D*_3_ were relatively accurate.

#### 4.2.2. Fitting Parameter D_4_

At the reference temperature (298 K), Equation (2) became
(27)$$\varepsilon_{f}=\left[D_{1}+D_{2}exp\left(D_{3}\sigma^{*}\right)\right]\left(1+D_{4}ln\frac{\dot{\varepsilon_{e,p}}}{\dot{\varepsilon_{0}}}\right)$$
where the number of parameters *D*_1_, *D*_2_, and *D*_3_ denoted the fitting results of the previous section. According to the quasi-static tensile test results under different strain rates, the failure parameter values *D*_4_ of the SLM and forged 15-5 PH steel were 0.0773 and 0.1006, respectively. The comparison results between the fitting curve obtained according to Equation (27) and the test data points are shown in Figure 24.

#### 4.2.3. Fitting Parameter D_5_

At the reference strain rate (1 × 10^−3^ s^−1^), Equation (2) became
(28)$$\varepsilon_{f}=\left[D_{1}+D_{2}exp\left(D_{3}\sigma^{*}\right)\right]\left[1+D_{5}\frac{T-T_{r}}{T_{m}-T_{r}}\right]$$
where the *D*_1_, *D*_2_, and *D*_3_ parameters used the fitting results from the previous section. According to the high temperature tensile test results, the failure parameter *D*_5_ values of the SLM and forged 15-5 PH steel were −0.4944 and 0.9736, respectively. The comparison results between the fitting curve obtained according to Equation (28) and the test data points are shown in Figure 25.

As shown in Figure 25, the fitting curve of the SLM-formed 15-5 PH steel showed a downward trend, while the fitting curve of the forged 15-5 PH steel showed an upward trend. This also confirmed that the failure strain of the SLM and forged 15-5 PH steel mentioned above was affected by temperature differently.

## 5. Model Verification

To verify the accuracy of the J-C constitutive models obtained above for the two types of steels, ballistic penetration experiments and the corresponding numerical simulations based on these constitutive models were carried out. The numerical results of projectile wear and penetration depths were compared with the corresponding experimental results.

### 5.1. Experimental and Numerical Methodologies

A 30-mm smooth bore gun was used as the launching platform, and the test layout is shown in Figure 26. To avoid ballistic deflection of the projectile during long-distance flight after it exited the muzzle, the muzzle was located about 3.5 m away from the target surface.

The test design speed was 800–1400 m/s, the body diameter was 14.5 mm, the length diameter ratio was 5, and the CRH was 4, as shown in Figure 27a,b. A 30-mm evenly divided 3-lobed missile carrier was designed by using sub-aperture launch technology, as shown in Figure 27c, where the size of the concrete target plate was Φ 450 × 350 mm, surrounded by 3-mm steel hoops. Figure 27d shows the physical object. We established the numerical simulation projectile and target plate with the same size as the test, as shown in Figure 28.

To reduce the calculation period, the target plate was processed in two steps. First, because in this work we studied forward penetration, we used a quarter model to model. Second, the mesh was densified within the range of 30 mm in the radius direction of the target point of the projectile. In the axial direction, a transition grid was used, and the first grid transition was conducted within the range of 30–100 mm from the axis, with a transition ratio of 4:2. A second grid transition was conducted within the range of 100–225 mm from the axis, with a transition ratio of 3:1. In the radial direction, the parts 30–100 mm away from the axis and 100–225 mm away from the axis were treated by gradual grid change, with a gradual change rate of 1.05, to ensure the correct propagation of stress waves in the target plate.

The projectile adopted the *MAT_JOHNSON_COOK material model, and the material parameters are shown in Table 12. The *MAT_RHT material model was adopted for concrete, and the material parameters are shown in Table 13.

### 5.2. Comparison between Numerical and Experimental Results

The measured impact velocities of the SLM shaped projectile in this experiment were 878, 943, and 1108 m/s, and the impact velocities of the forged projectile were 863, 949, and 1128 m/s. The damage results of the target plate when the impact speed of the SLM shaped projectile was 1108 m/s and the damage results of the target plate when the impact speed of the forged projectile was 1128 m/s were analyzed in cross-section. The results are shown in Figure 29.

As shown in Figure 29, when the penetration speed exceeded 1100 m/s, the 0.35-m concrete target plate was completely penetrated. To obtain clear and intuitive crack distribution, the target plate section was soaked with water. The water evaporation in the crack area was slower than that in the crack-free area, and the crack growth pattern with the water marks shown in the figure appeared. After the SLM-shaped projectile penetrated the concrete target, the open pit area of the target surface was small, the back pit area was large, the back of the target exhibited large collapse, and the internal crack of the target plate was small, but the crack was wide. After the forged projectile penetrated into the concrete target, the open pit area of the target plate was large, the back pit area was large, and many cracks were observed in the target plate, but the cracks were narrow.

The speed condition of the numerical simulation was consistent with the test. The wear characteristics of the projectile after the test and the numerical simulations are shown in Figure 30. The wear results of the projectile length are shown in Table 14, and the penetration depth results of the test and the numerical simulations are shown in Table 15.

As shown in Figure 30, under the penetration condition with a low target velocity, the surface of the projectile was more likely to stick to the concrete particles after the penetration test. After the penetration test, there were obvious striated scratches on the surface of the projectile; however, the numerical simulation failed to produce this phenomenon. This was due to the friction between the projectile and the concrete during the penetration test, which caused the surface of the projectile to form a high-temperature and high-pressure environment, and the surface was scratched by the concrete particles. In the numerical simulation, due to the limitation of grid size, similar scratches in the test did not form. With an increase in target velocity, the wear of the projectile head became more severe.

Table 14 shows the wear results of the projectile length. The error between the test and the numerical simulation was no greater than 2%, which further verified the reliability of the numerical model. With an increase in target velocity, the wear of the projectile length became more serious. The SLM-shaped projectiles were more wear-resistant than the forged projectiles, indicating that the SLM forming process had a better anti-wear effect on projectiles.

Table 15 shows the penetration results of the SLM and forged pellets into concrete, where the error between the test and numerical simulation was no greater than 10%. The reliability of the numerical model was verified by experiments. Because the penetration test could not guarantee the perfect vertical angle of impact, the penetration trajectory would deviate to different degrees; thus, the penetration depth was slightly larger than the thickness of the target plate. The penetration ability of the SLM-shaped projectiles was slightly higher than that of the forged projectiles, which indicated that the SLM forming process was helpful for improving the ability of the projectiles to penetrate into concrete. When the velocity of the warhead was greater than 1100 m/s, it completely penetrated the 350-mm-thick concrete.

## 6. Conclusions

In this work, quasi-static compression, SHPB, notch tensile, quasi-static tensile under different strain rates, high-temperature compression, and high-temperature tensile tests were carried out on the processed specimens of SLM and forged 15-5 PH steel. Parameters A, B, n, C, and m, as well as failure model parameters D_1_, D_2_, D_3_, D_4_, and D_5_ of the J-C model, for the SLM and forged 15-5 PH steel were fitted. The results showed that the mechanical properties of the SLM forming process were better than those of the forging process, and the yield strength of the specimen increased by 13.1% compared to the forged specimen. Through experiments and numerical simulations, the penetration ability of the projectile under the two processes was obtained. We found that the SLM shaped projectile had a stronger penetration ability and better abrasion resistance than the forged projectile. Through test verification, we found that the error of penetration depth of the test and numerical simulation results was within 10%, and the wear error of projectile length after the test and numerical simulations was not more than 2%, which indicated the reliability of the numerical simulation parameters.

## Figures and Tables

**Figure 1 materials-16-00800-f001:**
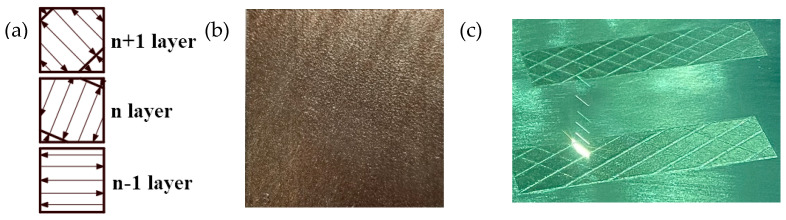
SLM forming scanning strategy diagram: (**a**) Scanning strategy (strip partition, partition width 20 mm, laser S-shaped two-way routing, interlayer rotation 67°); (**b**) SLM-shaped test piece; (**c**) SLM laser routing.

**Figure 2 materials-16-00800-f002:**
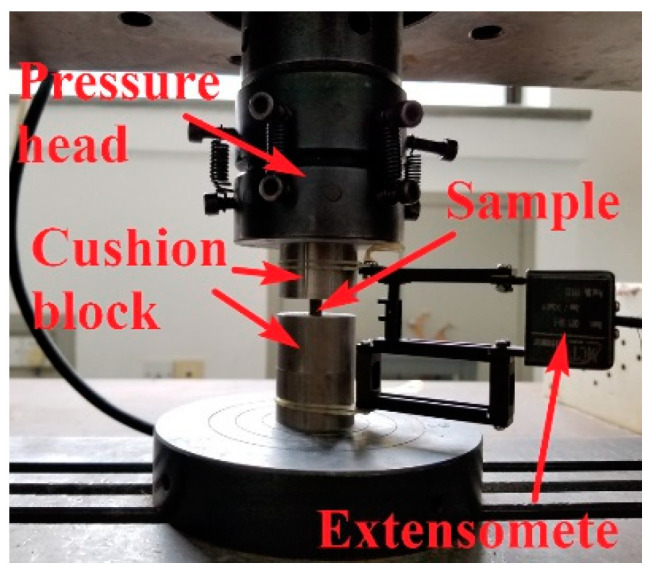
Quasi-static compression test equipment and specimen clamping diagram.

**Figure 3 materials-16-00800-f003:**
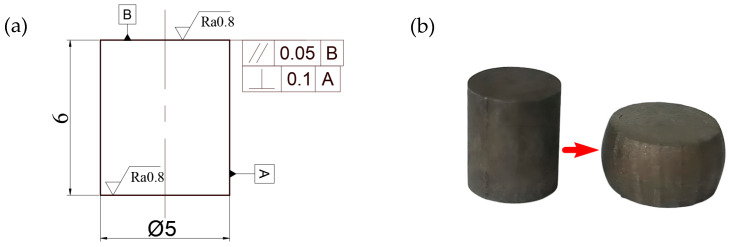
Compression comparison diagram of the test piece: (**a**) Compression specimen 2D engineering drawing, and (**b**) Before and after compression specimen test.

**Figure 4 materials-16-00800-f004:**
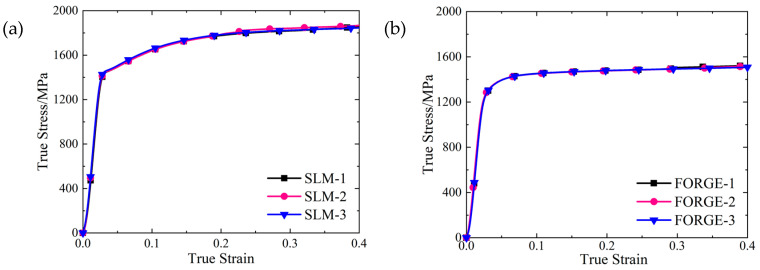
True stress-strain curves of the 15-5 PH steel: (**a**) SLM-shaped specimen, and (**b**) forged test piece.

**Figure 5 materials-16-00800-f005:**
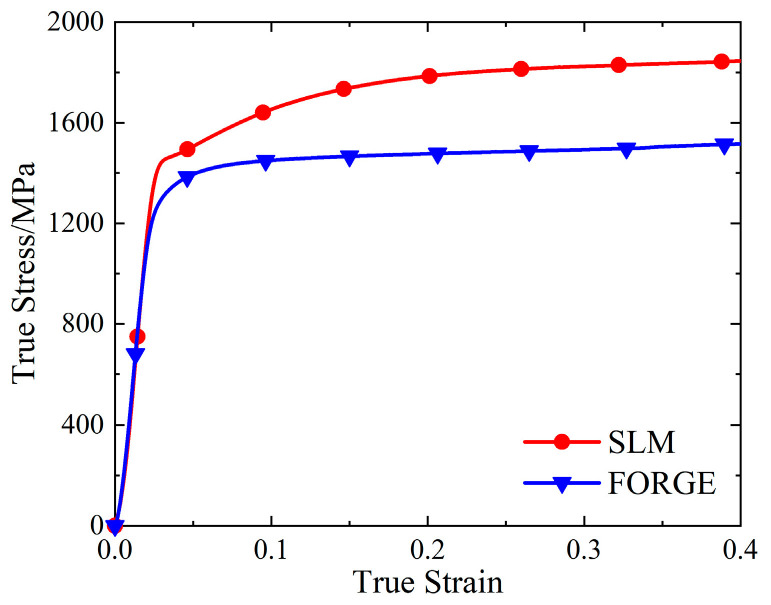
Comparison of the true stress-strain curves of 15-5 PH steel.

**Figure 6 materials-16-00800-f006:**
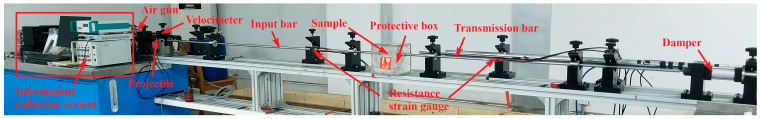
SHPB test device.

**Figure 7 materials-16-00800-f007:**
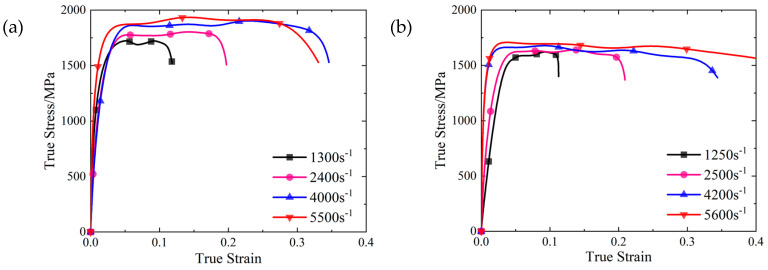
SHPB stress-strain curves of 15-5 PH steel: (**a**) the SLM-shaped specimen, and (**b**) the forged test piece.

**Figure 8 materials-16-00800-f008:**
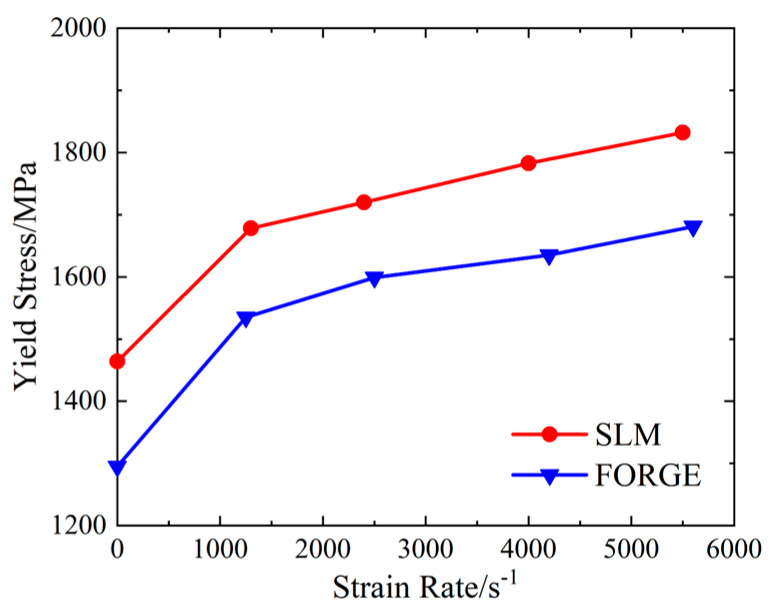
Relationship between the compressive yield strength and strain rate of 15-5 PH steel.

**Figure 9 materials-16-00800-f009:**
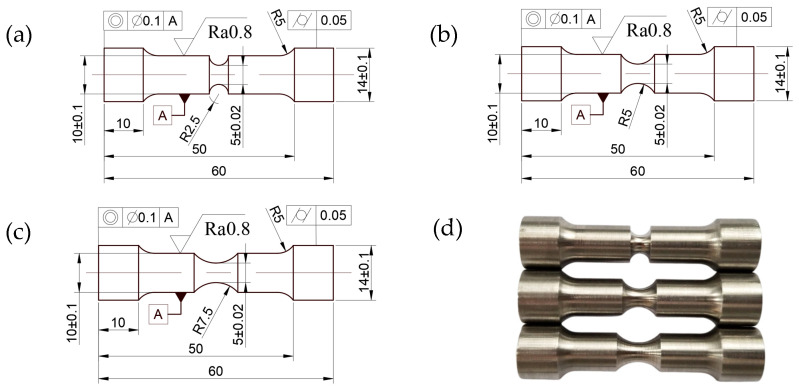
Dimensions and outline of the notch tensile test piece: (**a**–**c**) Tensile specimen 2D engineering drawing, and (**d**) Tensile samples.

**Figure 10 materials-16-00800-f010:**
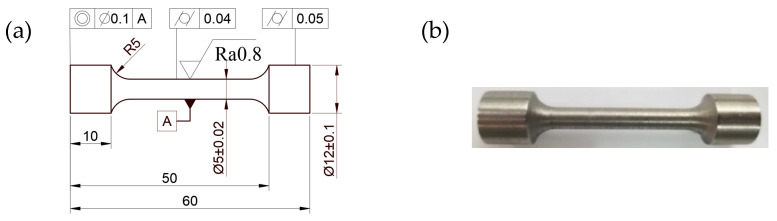
Dimensions and outline of the tensile test piece: (**a**) Tensile specimen 2D engineering drawing, and (**b**) Tensile sample.

**Figure 11 materials-16-00800-f011:**
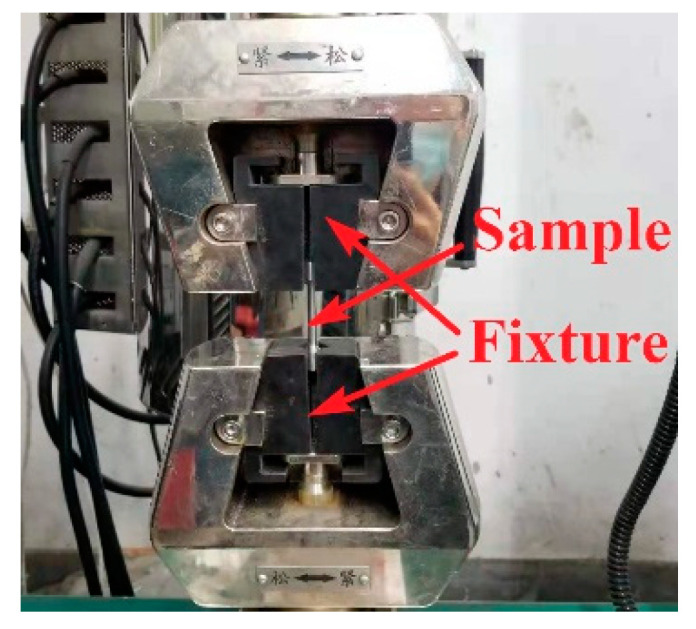
Clamping diagram of the cylindrical test piece.

**Figure 12 materials-16-00800-f012:**
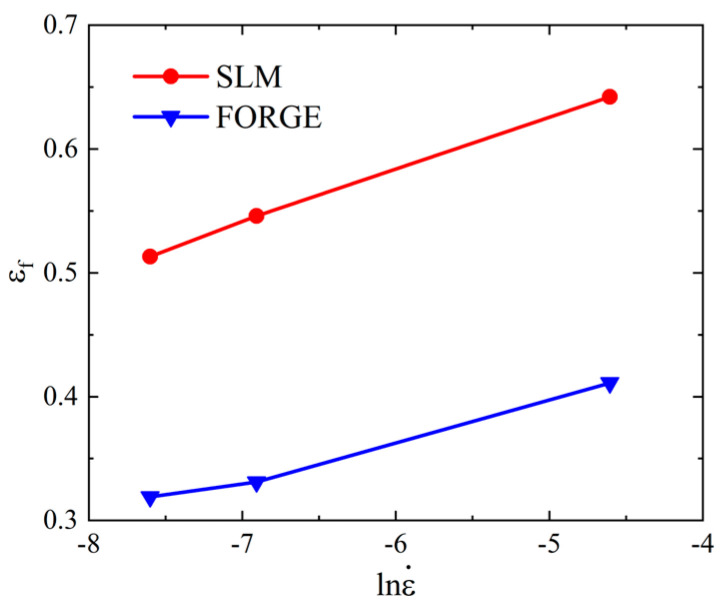
Relationship between the failure strain and strain rate under different working conditions.

**Figure 13 materials-16-00800-f013:**
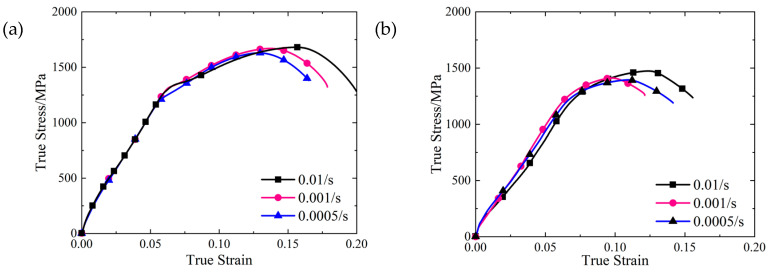
Tensile true stress-strain curves of the specimens under different working conditions: (**a**) SLM-shaped specimen and (**b**) forged test piece.

**Figure 14 materials-16-00800-f014:**
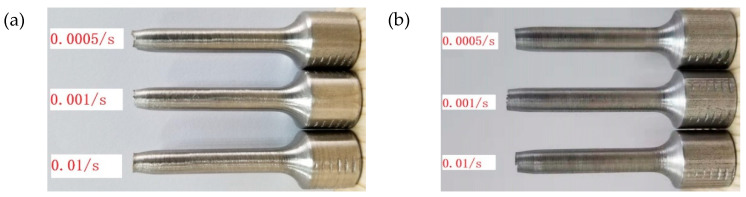
Fracture morphology of tensile test piece: (**a**) SLM-shaped test piece and (**b**) forged test piece.

**Figure 15 materials-16-00800-f015:**
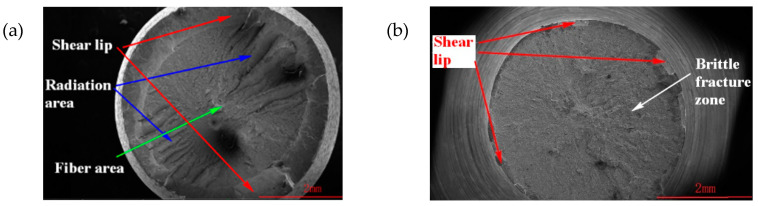
SEM comparison images of (**a**) SLM-shaped specimen, and (**b**) forged test piece.

**Figure 16 materials-16-00800-f016:**
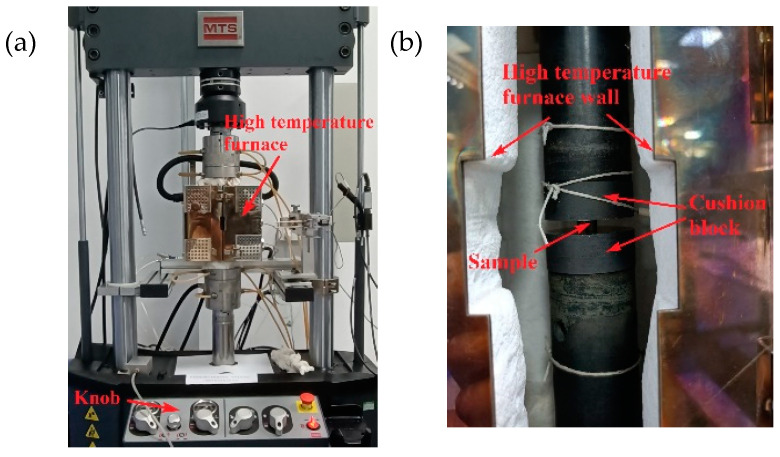
High-temperature compression test: (**a**) High-temperature compression test equipment, (**b**) Specimen clamping local diagram.

**Figure 17 materials-16-00800-f017:**
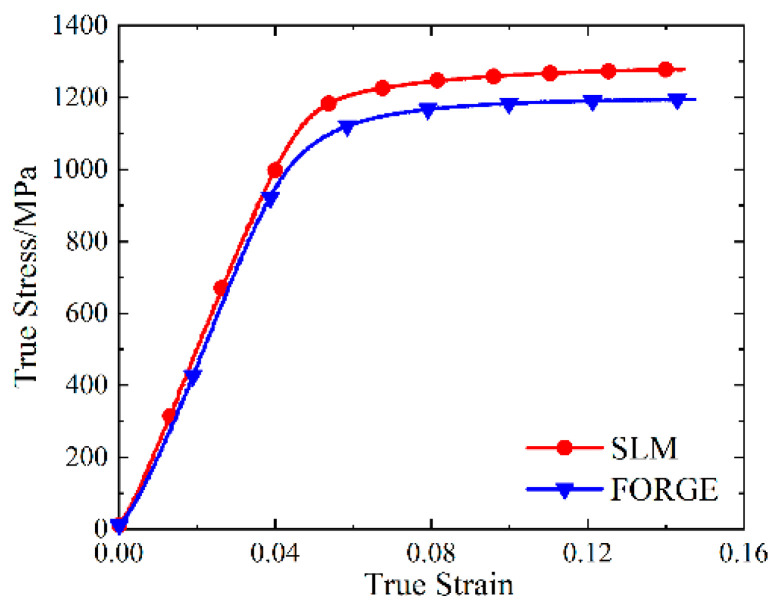
Real stress-strain curve of high-temperature compression at 400 °C.

**Figure 18 materials-16-00800-f018:**
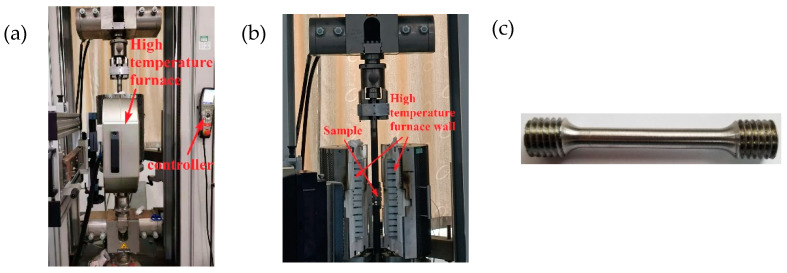
High-temperature tensile test: (**a**) High-temperature tensile test equipment, (**b**) internal layout of clamping, and (**c**) test piece.

**Figure 19 materials-16-00800-f019:**
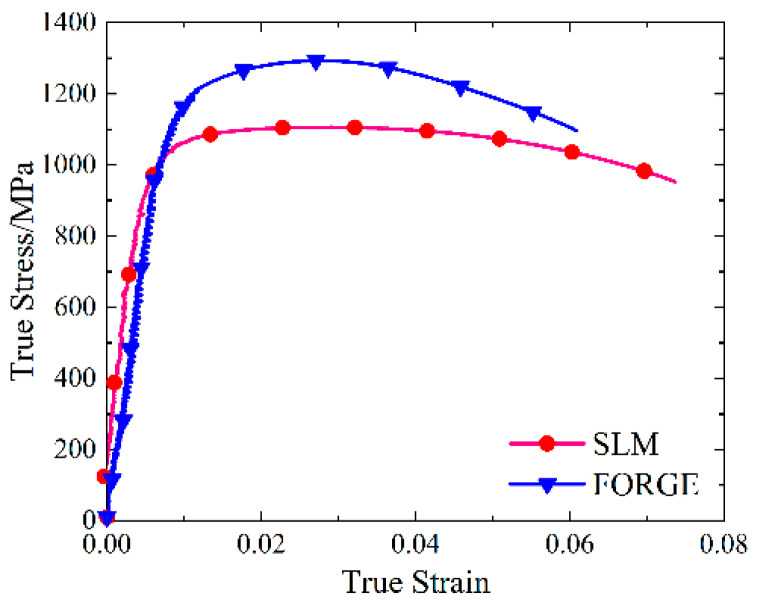
Tensile true stress-strain curves at 400 °C.

**Figure 20 materials-16-00800-f020:**
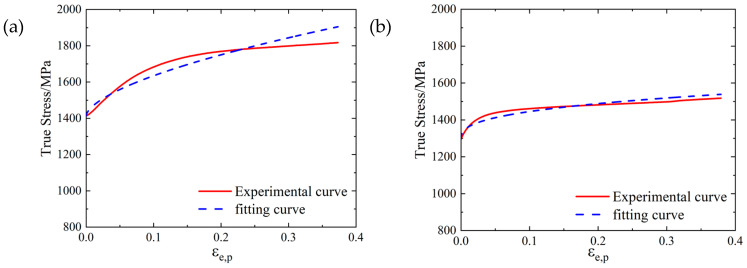
Fitting results of *A*, *B* and *n* for the 15-5 PH steel: (**a**) SLM-shaped test piece, (**b**) forged test piece.

**Figure 21 materials-16-00800-f021:**
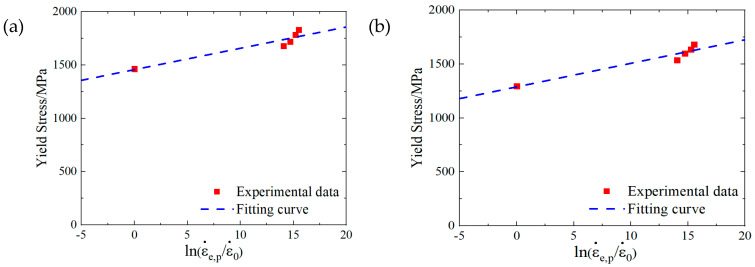
Fitting results of the 15-5 PH steel parameter C: (**a**) SLM-shaped specimen, and (**b**) forged test piece.

**Figure 22 materials-16-00800-f022:**
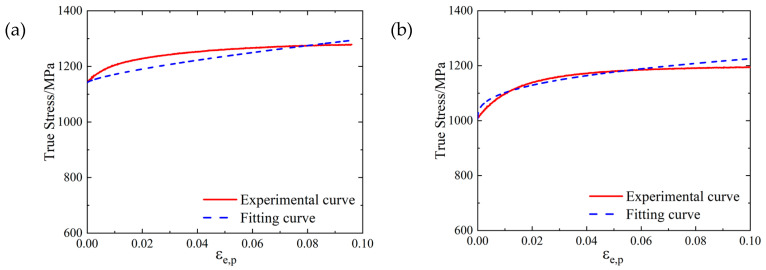
Fitting results of the 15-5 PH steel parameter *m*: (**a**) SLM-shaped test piece, (**b**) forged test piece.

**Figure 23 materials-16-00800-f023:**
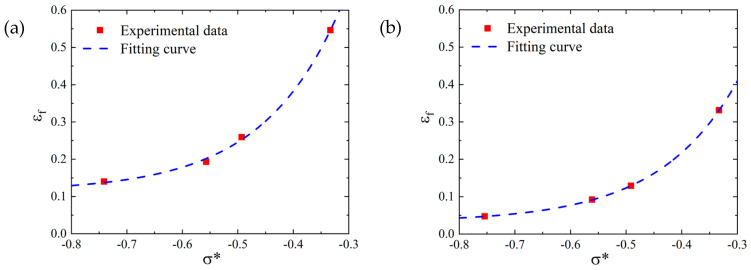
Fitting results of the 15-5 PH steel parameters *D*_1_, *D*_2_, and *D*_3_: (**a**) SLM-shaped test piece, and (**b**) forged test piece.

**Figure 24 materials-16-00800-f024:**
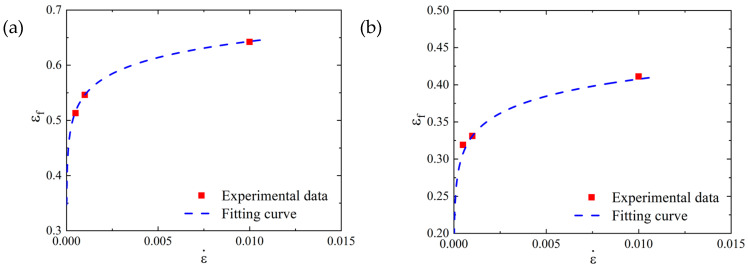
*D*_4_ fitting results of 15-5 PH steel parameters: (**a**) SLM-shaped specimen, and (**b**) forged test piece.

**Figure 25 materials-16-00800-f025:**
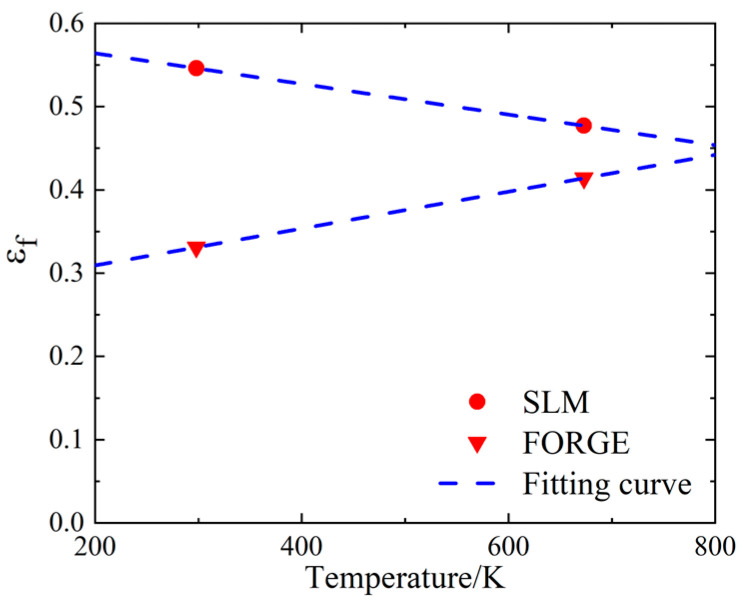
Fitting results of 15-5 PH steel parameter *D*_5_.

**Figure 26 materials-16-00800-f026:**
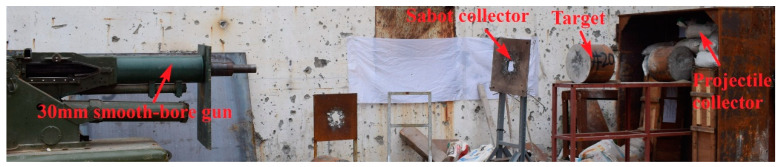
Site layout.

**Figure 27 materials-16-00800-f027:**
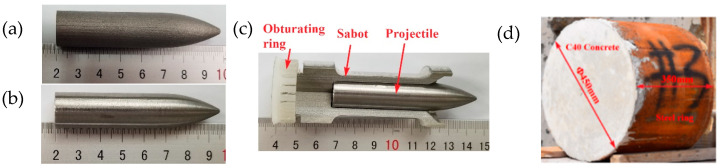
Test projectile and target plate: (**a**) schematic diagram of the projectile body; (**b**) SLM shaped projectile, (**c**) forged body, and (**d**) detachable cartridge holder.

**Figure 28 materials-16-00800-f028:**
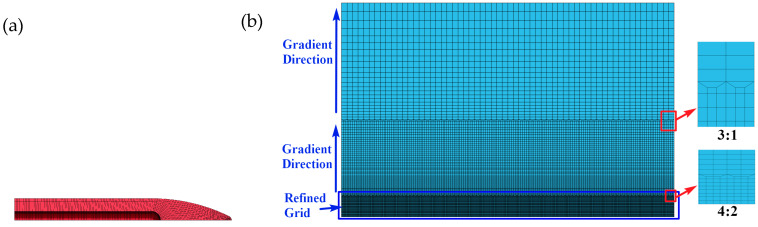
Establishment of the numerical model: (**a**) mesh division model of the projectile, and (**b**) mesh generation model of the target plate.

**Figure 29 materials-16-00800-f029:**
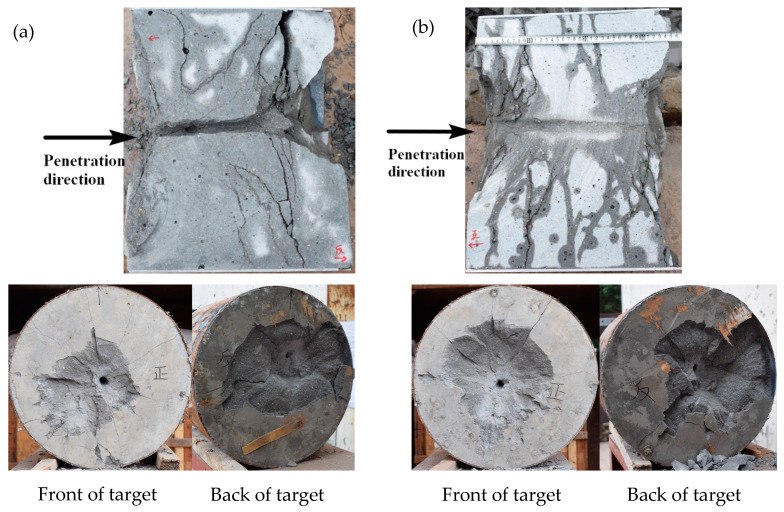
Damage of the target plate: (**a**) penetration results of the SLM shaped projectile at 1108 m/s, and (**b**) penetration results of the forged projectile at 1128 m/s.

**Figure 30 materials-16-00800-f030:**
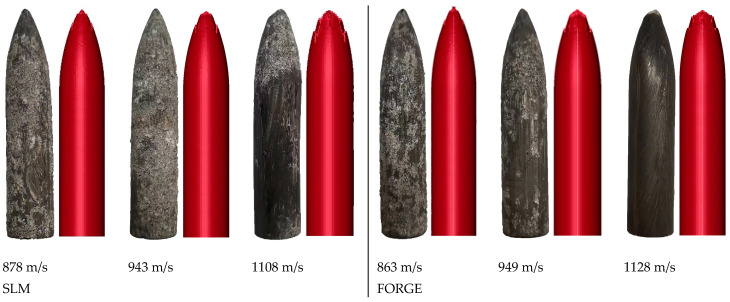
Comparison of projectile wear after penetration at different speeds.

**Table 1 materials-16-00800-t001:** Chemical composition of the forged 15-5 PH steel (wt%).

Cr	Ni	Cu	Mn	Si	C	P	Nb	S	Fe
14.0–15.5	3.5–5.5	2.5–4.5	1.0	1.0	0.07	0.04	0.15–0.45	≤0.03	Bal

**Table 2 materials-16-00800-t002:** Chemical composition of SLM-formed 15-5 PH steel powder (wt%).

Cr	Ni	Cu	Mn	Si	C	Mo	Nb	S	P	O	N	Fe
14.8	4.6	3.6	0.1	0.5	0.01	0.1	0.34	≤0.01	0.01	0.03	0.09	Bal

**Table 3 materials-16-00800-t003:** Granularity distribution of 15-5 PH steel powder.

Granularity (μm)	≤53	≤63
Percentage (%)	96.4	99.9

**Table 4 materials-16-00800-t004:** Dynamic yield strength of the 15-5 PH steel.

	Strain Rate (s^−1^)	Yield Strength (MPa)
SLM	1300	1678
	2400	1720
	4000	1783
	5500	1832
Forged	1250	1535
	2500	1599
	4200	1635
	5600	1681

**Table 5 materials-16-00800-t005:** Notch tensile test results of the SLM-formed 15-5 PH steel.

Notch Radius (mm)	Initial Section Diameter (mm)	Stress Triaxial Degree	Fracture Diameter (mm)	Failure Strain
2.5	5.03	0.741	4.69	0.140
5	5.01	0.557	4.55	0.193
7.5	5.19	0.493	4.56	0.259
∞	4.94	0.333	3.76	0.546

**Table 6 materials-16-00800-t006:** Notch tensile test results of the forged 15-5 PH steel.

Notch Radius (mm)	Initial Section Diameter (mm)	Stress Triaxial Degree	Fracture Diameter (mm)	Failure Strain
2.5	5.23	0.754	5.11	0.047
5	5.12	0.561	4.89	0.092
7.5	5.13	0.491	4.81	0.129
∞	4.98	0.333	4.22	0.331

**Table 7 materials-16-00800-t007:** Quasi-static tensile test results of the 15-5 PH steel under different working conditions.

Working Condition	Strain Rate (s^−1^)	Strain Rate Initial Diameter (mm)	Fracture Diameter (mm)	Elongation after Fracture (%)	Reduction of Area (%)	Stress Triaxial Degree	Failure Strain
SLM	5 × 10^−4^	5.04	3.90	12.1	40.1	0.333	0.513
	1 × 10^−3^	4.94	3.76	12.4	42.1	0.333	0.546
	1 × 10^−2^	5.1	3.7	14.4	47.4	0.333	0.642
FORGE	5 × 10^−4^	5.02	4.28	6.53	27.3	0.333	0.319
	1 × 10^−3^	4.98	4.22	6.93	28.2	0.333	0.331
	1 × 10^−2^	5.06	4.12	8.56	33.7	0.333	0.411

**Table 8 materials-16-00800-t008:** Tensile mechanical properties of 15-5 PH steel under different working conditions.

Working Condition	Strain Rate (s^−1^)	Tensile Strength (MPa)
SLM	5 × 10^−4^	1629
	1 × 10^−3^	1670
	1 × 10^−2^	1681
FORGE	5 × 10^−4^	1394
	1 × 10^−3^	1411
	1 × 10^−2^	1474

**Table 9 materials-16-00800-t009:** Failure strain of the high-temperature tensile test piece.

Working Condition	Strain Rate (s^−1^)	Initial Diameter (mm)	Fracture Diameter (mm)	Elongation after Fracture (%)	Reduction of Area (%)	Stress Triaxial Degree	Failure Strain
SLM	1 × 10^−3^	5.08	4.22	6.0	31.0	0.333	0.371
FORGE	1 × 10^−3^	4.92	4.00	5.5	33.9	0.333	0.414

**Table 10 materials-16-00800-t010:** J-C constitutive parameters *A*, *B*, and *n* of the SLM and forged 15-5 PH steel.

Working Condition	*A*/MPa	*B*/MPa	*n*
SLM	1464	1073	0.7286
FORGE	1295	346	0.3618

**Table 11 materials-16-00800-t011:** Parameters *D*_1_, *D*_2_, and *D*_3_ of the SLM and forged 15-5 PH steel.

Working Condition	*D* _1_	*D* _2_	*D* _3_
SLM	0.1136	4.6159	7.1074
FORGE	0.0322	3.1782	7.0999

**Table 12 materials-16-00800-t012:** Main parameters of the projectile materials.

Projectile	A/MPa	B/MPa	n	C	m	*D* _1_	*D* _2_	*D* _3_	*D* _4_	*D* _5_
SLM	1464	1073	0.7286	0.01368	1.1242	0.1136	4.6159	7.1074	0.0773	−0.4944
FORGE	1295	346	0.3618	0.01687	1.1230	0.0322	3.1782	7.0999	0.1006	0.9736

**Table 13 materials-16-00800-t013:** Main parameters of the concrete materials.

Name	ρ $/(g\cdot cm^{-3})$	$f_{c}/\mathrm{MPa}$	$f_{t}/\mathrm{MPa}$	$f_{t}{}^{*}$	G	$\varepsilon_{p}^{m}$	B	$D_{1}$	$D_{2}$
**Concrete**	2.23	40.8	1.41	0.1	10.5	0.01	0.0105	0.04	1

**Table 14 materials-16-00800-t014:** Projectile length wear data.

Projectile Type	Penetration Velocity (m/s)	Original Length of Projectile (mm)	Bullet Wear Length (mm)		
			Experiment	Numerical Simulation	Error (%)
SLM	878	72.5	7.14	7.09	0.70%
	943		7.08	7.00	1.13%
	1108		6.92	6.83	1.30%
FORGE	863	72.5	7.13	7.08	0.70%
	949		7.03	6.98	0.71%
	1128		6.91	6.78	1.88%

**Table 15 materials-16-00800-t015:** Penetration test results.

Projectile Type	Penetration Velocity (m/s)	Penetration Depth (mm)		
		Experiment	Numerical Simulation	Error (%)
SLM	878	289.4	261.7	9.57
	943	299.8	281.2	6.20
	1108	368.0 *	--	--
FORGE	863	275.3	252.2	8.39
	949	293.1	270.0	7.88
	1128	359.0 *	--	--

Notes: * represents a projectile penetrating a target.

## Data Availability

The data presented in this study are available on request from the corresponding author.

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
