# Peer review of "Determination of Johnson–Cook Constitutive of 15-5 PH Steel Processed by Selective Laser Melting"

_materials, 2023, doi:10.3390/ma16020800_

Round 1

Reviewer 1 Report

The materials constants used in the J-C constitutive equation for 15-5 PH steels; laser melting and conventional forging processes, are determined based on results obtained by great and various experiments on strain rate dependency, compressive and tensile states, triaxial conditions, temperature, etc. The verification of the determined J-C constitutive eq. was carried out by comparison between experimental and numerical simulation results on the concrete target penetration tests. As results of the experiments and the numerical simulations, it was concluded that the steel by the laser melting had the stronger penetration ability than that by the conventional forging, and that the numerical simulation using the determined J-C eq. indicates sufficient reliability. 

The comments are as followings:

1)    Figure 1 is not clear. The contrast of pictures should be better and add more detailed descriptions.

2)    The waist drum shape in Fig.3 seems to be large still. If the numerical simulation using the determined J-C const try to be compared with the final deformation, the reliability will be enhanced further.  

3)    In the case of impact phenomenon, impact energy is one of important parameters. As for Figures 7 and 8, not just strength but also strain is important. The impact energy absorption due to the plastic deformation should be mentioned.

4)    As for the description of L 405 to L 418, the ductile failure homologous criteria could be better to refer the following paper:

A homologous ductile failure criterion for generalized stress state, M. Futakawa, N. Butler, Engineering Fracture Mechanics, Vol 54, pp.349-359, 1996.

5)    What code was used for numerical simulation ?  Mention

6)    What is the definition of penetrate on the concrete plate, in particular in the simulation ?

7)    Indicate the penetrated shape of the target plate in the simulation, and compare with the pictures in Figure 29.

8)    As for the expression between line 670-674, the temperature due to the friction will be estimated in the simulation from the energy absorption relating the plastic deformation according to the contact interface between the projectile and the concrete.

9)    Small mistakes in spellings: L384   d -> b, L 501 Mpa -> MPa,

In Fig.20 fitting curve -> Fitting curve , In Fig. 21 experi -> Experi  fitting -> Fitting

L 556  d) -> b),  L 573 d) -> b), L582  d) -> b)   Please check in more detailed.

Author Response

Dear reviewer:

Thank you very much for your suggestions, they have helped me improve the quality of my paper. I have corrected the errors in my paper and followed your suggestions. These changes are shown in red font in the revised paper. In this letter, I have responded to your questions. Please let me know if I need to make additional changes to my paper.

Reviewer 2 Report

The paper deals with the determination of Johnson-Cook constitutive of 15-5 PH steel processed by selective laser melting

There are some major issues to be addressed by the authors before the paper is recommended for publication.

First of all, indicate somewhere in the paper what the post-processing treatments the specimens underwent after the SLM production.

What is the built direction of the specimens, especially those for the notch and quasi-static tensile tests? Did the authors analyse if different built directions could influence the mechanical properties? What is the original dimension of the specimens before they are machined up to the right dimensions for the tests?

How many specimens have been tested for retrieving each curve? Have the authors measured the specimens before and after the tests?

Please, enlarge the pictures in Figures 2, 3, 11, 16, and 18.

There are plenty of missing lines in the sketch of Figure 9. Please correct them.

Finally, the authors are asked to provide an explanation of why the SLM properties are so higher than the forged ones.

Author Response

Dear reviewer:

Thank you very much for your suggestions; they have helped me improve the quality of my paper. I have corrected the errors in my paper and followed your suggestions. These changes are shown in red font in the revised paper. In this letter, I have responded to your questions. Please let me know if I need to make additional changes to my paper.

Reviewer 3 Report

This paper deals with the constitutive properties of 15-5 PH steel processed by laser powder bed fusion. This is a very well-written paper containing numerous and valuable information on the properties of the new material. The conclusions are supported concretely by extensive experimental studies and discussions. Therefore the reviewer would like to recommend this paper to be accepted. There is just one minor point that can be considered by the authors to improve the quality of the paper further, as below:

It is recommended to use the standard term of the additive manufacturing process instead of selective laser melting (SLM). The ISO/ASTM standard of this process is laser-based powder bed fusion of metals (PBF-LB/M), according to ISO 52911-1 standard.

Author Response

(The authors gave the same response as above.)

Round 2

Reviewer 2 Report

No one of my comments have been addressed.

If you look at the author’s Response File, you can see that the authors have not replied sufficiently to my comments, and moreover, the revised manuscript has the same pictures as the original one. In particular, Fig. 9 still has some error with the lines missing.   Anyway, I suggest the authors highlight the modified parts in yellow along with the revised manuscript.

Author Response

(The authors gave the same response as above.)

Round 3

Reviewer 2 Report

There are some missing lines in Fig. 9. See in orange the missing lines of one of the three sketches.

Author Response

(The authors gave the same response as above.)
